# Glance and Focus: a Dynamic Approach to Reducing Spatial Redundancy in Image Classification

**Yulin Wang   Kangchen Lv   Rui Huang   Shiji Song   Le Yang   Gao Huang**[*]
Department of Automation, Tsinghua University, Beijing, China,
Beijing National Research Center for Information Science and Technology (BNRist),
`{wang-yl19, lkc17, huangr16, yangle15}@mails.tsinghua.edu.cn`,
`{shijis, gaohuang}@tsinghua.edu.cn`

## Abstract

The accuracy of deep convolutional neural networks (CNNs) generally improves when fueled with high resolution images. However, this often comes at a high computational cost and high memory footprint. Inspired by the fact that not all regions in an image are task-relevant, we propose a novel framework that performs efficient image classification by processing a sequence of relatively small inputs, which are strategically selected from the original image with reinforcement learning. Such a dynamic decision process naturally facilitates adaptive inference at test time, i.e., it can be terminated once the model is sufficiently confident about its prediction and thus avoids further redundant computation. Notably, our framework is general and flexible as it is compatible with most of the state-of-the-art light-weighted CNNs (such as MobileNets, EfficientNets and RegNets), which can be conveniently deployed as the backbone feature extractor. Experiments on ImageNet show that our method consistently improves the computational efficiency of a wide variety of deep models. For example, it further reduces the average latency of the highly efficient MobileNet-V3 on an iPhone XS Max by 20% without sacrificing accuracy. Code and pre-trained models are available at `https://github.com/blackfeather-wang/GFNet-Pytorch`.

## 1 Introduction

Modern convolutional networks (CNNs) are shown to benefit from training and inferring on high-resolution images. For example, state-of-the-art CNNs have achieved super-human-level performance on the competitive ILSVRC [9] competition with 224×224 or 320×320 images [47, 14, 61, 18, 22]. Recent works [50, 23] scale up the image resolution to 480×480 or even larger for higher accuracy. However, large images usually come at a high computational cost and high memory footprint, both of which grow quadratically with respect to the image height (or width) [38]. In real-world applications like content-based image search [54] or autonomous vehicles [3], computation usually translates into latency and power consumption, which should be minimized for both safety and economical reasons [17, 21, 43].

In this paper, we seek to reduce the computational cost introduced by high-resolution inputs in image classification tasks. Our motivation is that considerable *spatial redundancy* exists in the process of recognizing an image. In fact, CNNs are shown to be able to produce correct classification results with only a few class-discriminative patches, such as the head of a dog or the wings of a bird [38, 12, 8]. These regions are typically smaller than the whole image, and thus require much less computational resources. Therefore, if we can dynamically identify the "class-discriminative" regions of each individual image, and perform efficient inference only on these small inputs, then the

---

[*]Corresponding author.

spatial redundancy can be significantly reduced without sacrificing accuracy. To implement this idea, we need to address two challenges: 1) how to efficiently identify class-discriminative regions; and 2) how to adaptively allocate computation to each individual image, given that the number/size of discriminative regions may differ across different inputs.

In this paper, we present a two-stage framework, named *glance and focus*, to address the aforementioned issues. Specifically, the region selection operation is formulated as a sequential decision process, where at each step our model processes a relatively small input, producing a classification prediction with a confidence value as well as a region proposal for the next step. Each step can be done efficiently due to the reduced image size. For example, the computational cost of inferring a $96 \times 96$ image patch is only $18\%$ of that of processing the original $224 \times 224$ input. The whole sequential process starts with processing the full image in a down-sampled scale (*e.g.*, $96 \times 96$), serving as the initial step. We call it the *glance step*, at which the model produces a quick prediction of the input image using the *global* information. In practice, we find that a large portion of images with discriminative features can already be correctly classified with high confidence at the *glance step*, which is inline with the observation in [64]. When the *glance step* fails to produce sufficiently high confidence about its prediction, it will output a region proposal of the most discriminative region for the subsequent step to process. As the proposed region is usually a small patch of the original image with full resolution,

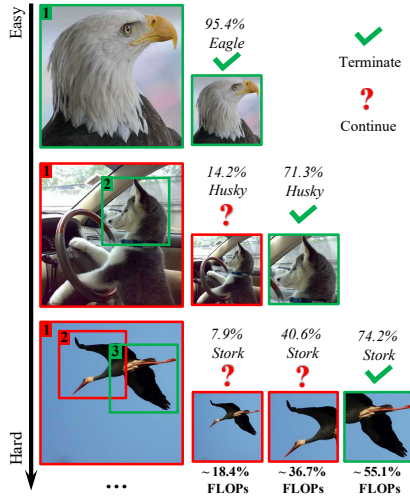

Figure 1: Examples for GFNet. "FLOPs" refers to the proportion of the computation required by GFNet (with 96×96 image patches) versus processing the entire 224×224 image.

we call these subsequent steps the *focus stage*. This stage proceeds progressively with iteratively localizing and processing the class-discriminative image regions, facilitating early termination in an adaptive manner, *i.e.,* the decision process can be interrupted dynamically conditioned on each input image. As shown in Figure 1, our method allocates computation unevenly across different images at test time, leading to a significant improvement of the overall efficiency. We refer to our method as *Glance and Focus Network (GFNet)*.

It is worth noting that the proposed GFNet is designed as a general framework, where the classifier and the region proposal network are treated as two independent modules. Therefore, most of the state-of-the-art light-weighted CNNs, such as MobileNets [17, 43, 16], CondenseNet [21], ShuffleNets [66, 36] and EfficientNet [50], can be deployed for higher efficiency. This differentiates our method from early recurrent attention methods [38] which adopt pure recurrent models. In addition, we focus on improving the computational efficiency under the adaptive inference setting, while most existing works aim to improve accuracy with fixed sequence length.

Besides its high computational efficiency, the proposed GFNet is appealing in several other aspects. For example, the memory consumption can be significantly reduced, and it is independent of the original image resolution as long as we fix the size of the region. Moreover, the computational cost of GFNet can be adjusted online without additional training (by simply adjusting the termination criterion). This enables GFNet to make full use of all available computational resources flexibly or achieve the required performance with minimal power consumption – a practical requirement of many real-world applications such as search engines and mobile apps.

We evaluate the performance of GFNet on ImageNet [9] with various efficient CNNs (*e.g.*, MobileNet-V3 [16], RegNet [40], EfficientNet [50], etc.) in the budgeted batch classification setting [20], where the test set comes with a given computational budget, and the anytime prediction setting [13, 20], where the network can be forced to output a prediction at any given point in time. We also benchmark the average latency of GFNet on an iPhone XS Max. Experimental results show that GFNet effectively improves the efficiency of state-of-the-art networks both theoretically and empirically. For example, when the MobileNets-V3 and ResNets are used as the backbone network, GFNet has up to $1.4\times$ and $3\times$ less Multiply-Add operations compared to the original models when achieving the same level of accuracy, respectively. Notably, the actual speedup on an iPhone XS Max (measured by average latency) is $1.3\times$ and $2.9\times$, respectively.

## 2    Related Work

**Computationally efficient networks.** Modern CNNs usually require a large number of computational resources. To this end, many research works focus on reducing the inference cost of the networks. A promising direction is to develop efficient network architectures, such as MobileNets [17, 43, 16], CondenseNet [21], ShuffleNets [66, 36] and EfficientNet [50]. Since deep networks typically have a considerable number of redundant weights [11], some other approaches focus on pruning [30, 31, 34, 35] or quantizing the weights [41, 24, 25]. Another technique is knowledge distillation [15], which trains a small network to reproduce the prediction of a large model. Our method is orthogonal to the aforementioned approaches, and can be combined with them to further improve the efficiency.

A number of recent works improve the efficiency of CNNs by *adaptively* changing the architecture of the network. For example, MSDNet [20] and its variants [32, 64] introduce a multi-scale architecture with multiple classifiers that enables it to adopt small networks for easy samples while switch to large models for hard ones. Another approach is to ensemble multiple models, and selectively execute a subset of them in the cascading [4] or mixing [45, 42] paradigm. Some other works propose to dynamically skip unnecessary layers [52, 55, 60] or channels [33].

**Spatial redundancy.** Recent research has revealed that considerable spatial redundancy occurs when inferring CNNs [10, 62]. Several approaches have been proposed to reduce the redundant computation in the spatial dimension. The OctConv [6] reduces the spatial resolution by using low-frequency features. The Spatially Adaptive Computation Time (SACT) [10] dynamically adjusts the number of executed layers for different image regions. The methods proposed in [19] and [62] skip the computation on some less important regions of feature maps. These works mainly reduce the spatial redundancy by modifying convolutional layers, while we propose to process the image in a sequential manner. Our method is general as it does not require altering the CNN architecture.

**Visual attention models.** Our GFNet is related to the visual attention models, which are similar to the human perception in that human usually pay attention to parts of the environment to perform recognition. Many existing works integrate the attention mechanism into image processing systems, especially in language-related tasks. For example, in image captaining and visual question answering, models are trained to concentrate on the related regions of the image when generating the word sequence [63, 53, 51, 27, 65]. In the context of image recognition, the attention mechanism is typically exploited to extract information from some task-relevant regions [29, 2, 26, 12, 8].

One similar work to our GFNet is the recurrent visual attention model proposed in [38]. However, our method differs from it in two important aspects: 1) we adopt a flexible and general CNN-based framework that is compatible with a wide variety of CNNs to achieve SOTA computational efficiency, instead of sticking to a pure RNN model; and 2) our network focuses on performing adaptive inference for higher efficiency, and the recurrent process can be terminated conditioned on each input. With these design innovations, the proposed GFNet has achieved new SOTA performance on ImageNet in terms of both the theoretical computational efficiency and actual inference speed. In addition, [46] shares a similar spirit to us in selecting important features with reinforcement learning, but it is not based on CNNs nor image data.

## 3    Method

In this section, we introduce the details of our method. As aforementioned, CNNs are capable of producing accurate image classification results with certain "class-discriminative" image regions, such as the face of a dog or the wings of a bird. Inspired by this observation, we propose a GFNet framework, aiming to improve the computational efficiency of CNNs by performing computation on the minimal image regions to obtain a reliable prediction. To be specific, GFNet allocates computation adaptively and progressively to different areas of an image according to their potential contributions to the final classification, and the process is terminated once the network is adequately confident.

### 3.1    Overview

In this subsection, we give an overview of the proposed GFNet (as shown in Figure 2). Details of its components will be presented in Section 3.2.

Given an image $x$ with the size $H \times W$, our method processes it with a sequence of $H' \times W'$ smaller inputs $\{\tilde{x}_1, \tilde{x}_2, \dots\}$, where $H' < H, W' < W$. These inputs are image patches directly cropped from certain locations of the image (except for $\tilde{x}_1$, which will be described later). The specific location of each patch is dynamically determined by the network using the information of all previous inputs.

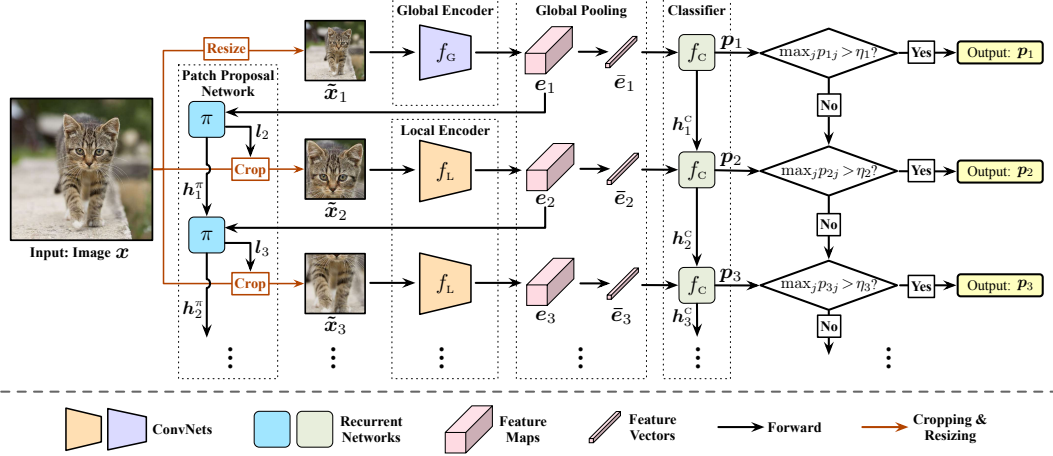

Figure 2: An overview of GFNet. Given an input image $x$, the model iteratively processes a sequence of patches $\{\tilde{x}_1, \tilde{x}_2, \dots\}$. The first input $\tilde{x}_1$ is a low-resolution version of $x$, and it is processed by the global encoder $f_G$ (*Glance Step*). The following ones $\{\tilde{x}_2, \dots\}$ are high-resolution patches cropped from $x$, which are fed into the local encoder $f_L$ (*Focus Stage*). At each step, GFNet produces a prediction with a classifier $f_C$, as well as decides the location of the next image patch using a patch proposal network $\pi$. This sequential decision process is terminated once sufficient confidence is obtained.

Ideally, the contributions of the inputs to classification should be descending in the sequence, such that the computational resources are first spent on the most valuable regions. However, given an arbitrary image, we do not have any specific prior knowledge on which regions are more important when generating the first patch $\tilde{x}_1$. Therefore, we simply resize the original image $x$ to $H' \times W'$ as $\tilde{x}_1$, which not only avoids the risk of wasting computation on less important regions caused by randomly localizing the initial region, but also provides necessary global information that is beneficial for determining the locations of the following patches.

**Inference.** We first describe the inference procedure of GFNet. At the $t^{\text{th}}$ step of the dynamic decision process, the CNN backbone ($f_G$ or $f_L$) receives the input $\tilde{x}_t$, and the model produces a softmax prediction $p_t$. Then the largest entry of $p_t$, *i.e.*, $\max_j p_{tj}$, (treated as confidence following earlier works [20, 64]) is compared to a pre-defined threshold $\eta_t$. If $\max_j p_{tj} > \eta_t$, then the sequential process halts, and $p_t$ will be outputted as the final prediction. Otherwise, the location of the next image patch $\tilde{x}_{t+1}$ will be decided, and $\tilde{x}_{t+1}$ will be cropped from the image as the input at the $(t+1)^{\text{th}}$ step. Note that the prediction $p_t$ and the location of $\tilde{x}_{t+1}$ are obtained using two recurrent networks, such that they exploit the information of all previous inputs $\{\tilde{x}_1, \tilde{x}_2, \dots, \tilde{x}_t\}$. The maximum length of the input sequence is restricted to $T$ by setting $\eta_T = 0$, while other confidence thresholds $\eta_t (1 \leq t \leq T-1)$ are determined according to the practical requirements for a given computational budget. The details of obtaining these thresholds are presented in Section 3.4.

**Training.** During training, we inactivate early-terminating by setting $\eta_t = 1 (1 \leq t \leq T - 1)$, and enforce all prediction $p_t$'s $(1 \leq t \leq T)$ to be correct with high confidence. For patch localization, we train the network to select the patches that maximize the increments of the softmax prediction on the ground truth labels between adjacent two steps. In other words, we always hope to find the most class-discriminative image patches that have not been seen by the network. This procedure exploits a policy gradient algorithm to address the non-differentiability.

### 3.2 The GFNet Architecture

The proposed GFNet consists of four components: a global encoder $f_G$, a local encoder $f_L$, a classifier $f_C$ and a patch proposal network $\pi$.

**Global encoder $f_G$ and local encoder $f_L$** are both deep CNNs that we utilize to extract deep representations from the inputs. They share the same network architecture but with different parameters. The former is applied to the resized original image $\tilde{x}_1$, while the later is applied to the selected image patches. We use two networks instead of one because we find that there is a discrepancy between the low-resolution inputs $\tilde{x}_1$ and the high-resolution local patches, which leads to degraded performance with a single encoder (detailed results are given in Section 4.3).

**Classifier** $f_\text{C}$ is a recurrent network that aggregates the information from all previous inputs and produces a prediction at each step. We assume that the $t^\text{th}$ input $\tilde{\boldsymbol{x}}_t$ is fed into the encoder, which obtains the corresponding feature maps $\boldsymbol{e}_t$. We perform the global average pooling on $\boldsymbol{e}_t$ to get a feature vector $\bar{\boldsymbol{e}}_t$, and produce the prediction $\boldsymbol{p}_t$ by $\boldsymbol{p}_t = f_\text{C}(\bar{\boldsymbol{e}}_t, \boldsymbol{h}_{t-1}^\text{C})$, where $\boldsymbol{h}_{t-1}^\text{C}$ is the hidden state of $f_\text{C}$, which is updated at the $(t-1)^\text{th}$ step. Note that it is unnecessary to maintain the feature maps for the *classifier* as classification usually does not rely on the spatial information they contain. The recurrent classifier $f_\text{C}$ and the aforementioned two encoders $f_\text{G}, f_\text{L}$ are trained simultaneously with the following classification loss:

$$\mathcal{L}_\text{cls} = \frac{1}{|\mathcal{D}_\text{train}|} \sum_{(\boldsymbol{x},y)\in\mathcal{D}_\text{train}} \left[ \frac{1}{T} \sum_{t=1}^{T} L_\text{CE}(\boldsymbol{p}_t, y) \right]. \tag{1}$$

Herein, $\mathcal{D}_\text{train}$ is the training set, $y$ denotes the label corresponding to $\boldsymbol{x}$ and $T$ is the maximum length of the input sequence. We use the standard cross-entropy loss function $L_\text{CE}(\cdot)$ during training.

**Patch proposal network** $\pi$ is another recurrent network that determines the location of each image patch. Given that the outputs of $\pi$ are used for the non-differentiable cropping operation, we model $\pi$ as an agent and train it using the policy gradient method. In specific, it receives the feature maps $\boldsymbol{e}_t$ of $\tilde{\boldsymbol{x}}_t$ at $t^\text{th}$ step, and chooses a localization action $\boldsymbol{l}_{t+1}$ stochastically from a distribution parameterized by $\pi$: $\boldsymbol{l}_{t+1} \sim \pi(\boldsymbol{l}_{t+1}|\boldsymbol{e}_t, \boldsymbol{h}_{t-1}^\pi)$, where $\boldsymbol{l}_{t+1} \in [0,1]^2$ is formulated as the normalized coordinates of the centre of the next patch $\tilde{\boldsymbol{x}}_{t+1}$. Here we use a Gaussian distribution during training, whose mean is outputted by $\pi$ and standard deviation is predefined as a hyper-parameter. At test time, we simply adopt the mean value as $\boldsymbol{l}_{t+1}$ for a deterministic inference process. We denote the hidden state maintained within $\pi$ by $\boldsymbol{h}_{t-1}^\pi$, which aggregates the information of all past feature maps $\{\boldsymbol{e}_1, \ldots, \boldsymbol{e}_{t-1}\}$. Note that we do not perform any pooling on $\boldsymbol{e}_t$, since the spatial information in the feature maps is essential for localizing the discriminative regions. On the other hand, we save the computational cost by reducing the number of feature channels

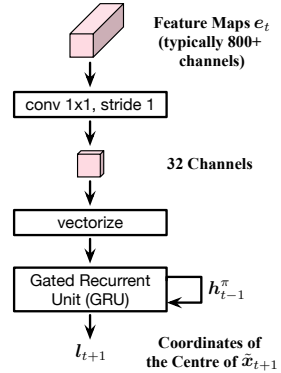

Figure 3: The architecture of the patch proposal network $\pi$.

using a $1\times1$ convolution. Such a design abandons parts of the information that are valuable for classification but unnecessary for localization. The architecture of $\pi$ is shown in Figure 3.

During training, after obtaining $\boldsymbol{l}_{t+1}$, we crop the $H'\times W'$ area centred at $\boldsymbol{l}_{t+1}$ from the original image $\boldsymbol{x}$ as the next input $\tilde{\boldsymbol{x}}_{t+1}$, and feed it into the network to produce the prediction $\boldsymbol{p}_{t+1}$. Then the patch proposal network $\pi$ receives a reward $r_{t+1}$ for the action $\boldsymbol{l}_{t+1}$, which is defined as the increments of the softmax prediction probability on the ground truth labels, *i.e.*, $r_{t+1} = p_{(t+1)y} - p_{ty}$, where $y \in \{1, \ldots, C\}$ is the label of $\boldsymbol{x}$ among $C$ classes. The goal of $\pi$ is to maximize the sum of the discounted rewards:

$$\max_\pi \mathbb{E}\left[ \sum_{t=2}^{T} \gamma^{t-2} r_t \right], \tag{2}$$

where $\gamma \in (0,1)$ is a pre-defined discount factor. Intuitively, through Eq. (2), we enforce $\pi$ to select the patches that enable the network to produce correct predictions in high confidence with as fewer patches as possible. In essence, we train $\pi$ to predict the location of the most beneficial region for the image classification at each step. Note that this procedure considers the previous inputs as well, since we compute the "increments" of the prediction probability.

### 3.3 Training Strategy

To ensure GFNet is trained properly, we propose a 3-stage training scheme, where the first two stages are indispensable, and the third stage is designed to further improve the performance.

**Stage I:** At first, we do not integrate the patch proposal network $\pi$ into GFNet. Instead, we randomly crop the patch at each step with a uniform distribution over the entire input image, and train $f_\text{G}$, $f_\text{L}$ and $f_\text{C}$ to minimize the classification loss $\mathcal{L}_\text{cls}$ (Eq. (1)). In this stage, the network is trained to adapt to arbitrary input sequences.

**Stage II:** We fix the two encoders and the classifier obtained from Stage I, and evoke a randomly initialized patch proposal network $\pi$ to decide the locations of image patches. Then we train $\pi$ using a policy gradient algorithm to maximize the total reward (Eq. (2)).

**Stage III:** Finally, we fine-tune the two encoders and the classifier with the fixed $\pi$ obtained from Stage II to improve the performance of GFNet with the learned patch selection policy.

### 3.4 Implementation Details

**Initialization of $f_G$ and $f_L$.** We initialize the local encoder $f_L$ using the ImageNet pre-trained models. Since the global encoder $f_G$ processes the resized image with lower resolution, we first fine-tune the ImageNet pre-trained models with all training samples resized to $H' \times W'$, and then initialize $f_G$ with the fine-tuned parameters. It is interesting that simply fine-tuning the pre-trained models using low-resolution images contributes to more efficient networks, which is discussed in Appendix B.1.

**Recurrent networks.** We adopt the gated recurrent unit (GRU) [7] in the classifier $f_C$ and the patch proposal network $\pi$. For MobileNets-V3 and EfficientNets, we use a cascade of fully-connected layers for more efficient implementation. The details are deferred to Appendix A.1.

**Regularizing CNNs.** In our implementation, we add a regularization term to Eq. (1), aiming to keep the capability of the two CNNs to learn linearly separable representations, namely

$$\mathcal{L}'_{\text{cls}} = \frac{1}{|\mathcal{D}_{\text{train}}|} \sum\nolimits_{(\boldsymbol{x},y) \in \mathcal{D}_{\text{train}}} \left\{ \frac{1}{T} \sum\nolimits_{t=1}^{T} \left[ L_{\text{CE}}(\boldsymbol{p}_t, y) + \lambda L_{\text{CE}}(\text{Softmax}(\text{FC}_t(\bar{\boldsymbol{e}}_t)), y) \right] \right\}, \quad (3)$$

where $\lambda > 0$ is a pre-defined coefficient. Herein, we define a fully-connected layer $\text{FC}_t(\cdot)$ for each step, and compute the softmax cross-entropy loss over the feature vector $\bar{\boldsymbol{e}}_t$ with $\text{FC}_t$. Note that, when minimizing Eq. (1), the two encoders are not directly supervised since all gradients flow through the classifier $f_C$, while Eq. (3) explicitly enforces a linearized deep feature space.

**Confidence thresholds.** One of the prominent advantages of GFNet is that both its computational cost and its inference latency can be tuned online according to the practical requirements via changing the confidence thresholds. Assume that the probability of obtaining the final prediction at $t^{\text{th}}$ step is $q_t$, and the corresponding computational cost or latency is $C_t$. Then the average cost for each sample can be computed as $\sum_t q_t C_t$. In the context of *budgeted batch classification* [20], the model needs to classify a set of samples $\mathcal{D}_{\text{test}}$ within a given computational budget $B > 0$, leading to the constraint $|\mathcal{D}_{\text{test}}| \sum_t q_t C_t \leq B$. We can solve this constraint for a proper $q_t$ and determine the threshold $\eta_t$ on the validation set. In our implementation, following [20], we let $q_t = z(1-q)^{t-1}q$, where $z$ is a normalizing constant to ensure $\sum_t q_t = 1$, and $0 < q < 1$ is an exit probability to be solved.

**Policy gradient algorithm.** We implement the proximal policy optimization (PPO) algorithm proposed by [44] to train the patch proposal network $\pi$. The details are introduced in Appendix A.2.

## 4 Experiments

In this section, we empirically evaluate the effectiveness of the proposed GFNet and give ablation studies. Code and pre-trained models are available at `https://github.com/blackfeather-wang/GFNet-Pytorch`.

**Dataset.** ImageNet is a 1,000-class dataset from ILSVRC2012 [9], with 1.2 million images for training and 50,000 images for validation. We adopt the same data augmentation and pre-processing configurations as [22, 14, 57]. In our implementation, we estimate the confidence thresholds of GFNet on the training set, since we find that it achieves nearly the same performance as estimating the thresholds on an additional validation set split from the training images.

**Setup.** We consider two settings to evaluate our method: (1) *budgeted batch classification* [20], where the network needs to classify a set of test samples within a given computational budget; (2) *anytime prediction* [13, 20], where the network can be forced to output a prediction at any given point in time. As discussed in [20], these two settings are ubiquitous in many real-world applications. For (1), we estimate the confidence thresholds to perform the adaptive inference as introduced in Section 3.4, while for (2), we assume the length of the input sequence is the same for all test samples.

**Networks.** The GFNet is implemented on the basis of several state-of-the-art CNNs, including MobileNet-V3 [16], RegNet-Y [40], EfficientNet [50], ResNet [14] and DenseNet [22]. These networks serve as the two deep encoders in our methods. In *Budgeted batch classification*, for each sort of CNNs, we fix the maximum length of the input sequence $T$, and change the model size (width, depth or both) or the patch size $(H', W')$ to obtain networks that cover different ranges of computational budgets. Note that we always let $H' = W'$. Details on both the network configurations and the training hyper-parameters are presented in Appendix A.3. We also compare GFNet with a number of highly competitive baselines, i.e., MnasNets [49], ShuffleNets-V2 [36], MobileNets-V2 [43], CondenseNets [21], FBNets [59], ProxylessNAS [5], SkipNet [55], SACT [10], GoogLeNet [48] and MSDNet [20].

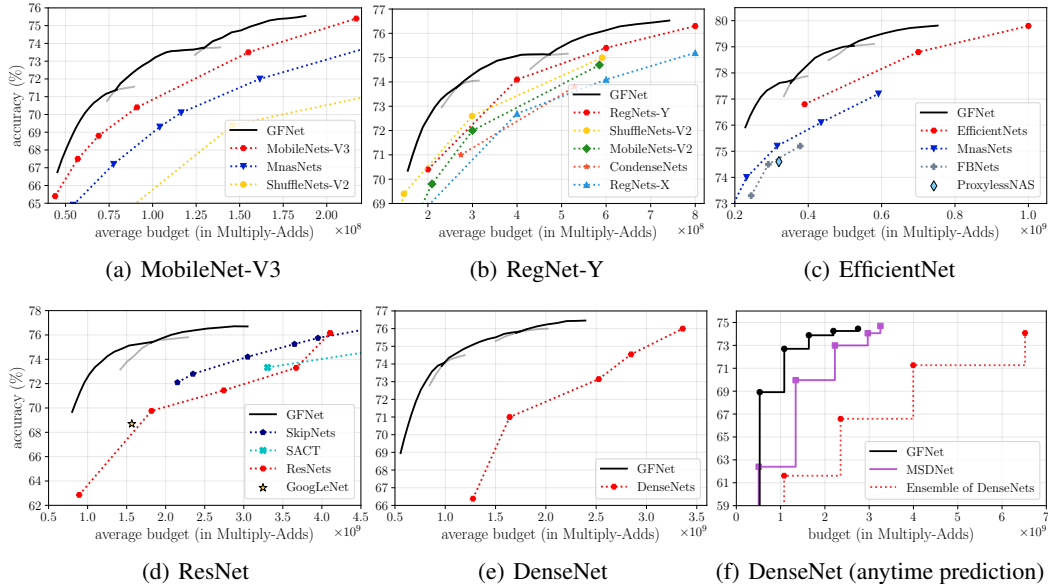

| (a) MobileNet-V3 | (b) RegNet-Y | (c) EfficientNet |
| --- | --- | --- |
| (d) ResNet | (e) DenseNet | (f) DenseNet (anytime prediction) |

Figure 4: Top-1 accuracy v.s. Multiply-Adds on ImageNet. The proposed GFNet framework is implemented on top of state-of-the-art efficient networks. Figures (a-e) present the results of *Budgeted batch classification*, while Figures (f) shows the *Anytime prediction* results.

## 4.1 Main Results

**Budgeted batch classification** results are shown in Figure 4 (a-e). We first plot the performance of each GFNet in a gray curve, and then plot the best validation accuracy with each budget as a black curve. It can be observed that GFNet significantly improves the performance of even the state-of-the-art efficient models with the same amount of computation. For example, with an average budget of $7 \times 10^7$ Multiply-Adds, the GFNet based on MobileNet-V3 achieves a Top-1 validation accuracy of $\sim 71\%$, which outperforms the vanilla MobileNet-V3 by $\sim 2\%$. With EfficientNets, GFNet generally has $\sim 1.4\times$ less computation compared with baselines when achieving the same performance. With ResNets and DenseNets, GFNet reduces the number of required Multiply-Adds for the given test accuracy by approximately $2 - 3\times$ times. Moreover, the computational cost of our method can be tuned precisely to achieve the best possible performance with a given budget.

**Anytime prediction.** We compare our method with another adaptive inference architecture, MSDNet [20], under the *Anytime prediction* setting in Figure 4 (f), where the GFNet is based on a DenseNet-121. For fair comparison, here we hold out 50,000 training images, following [20] (we do not do so in Figure 4 (e), and hence the *Budgeted batch classification* comparisons between GFNet and MSDNet are deferred to Appendix B.2). We also include the results of an ensemble of DenseNets with varying depth [20]. The plot shows that GFNet achieves $\sim 4 - 10\%$ higher accuracy than MSDNet when the budget ranges from $5 \times 10^8$ to $2.2 \times 10^9$ Multiply-Adds.

**Experiments on an iPhone.** A suitable platform for the implementation of GFNet may be mobile applications, where the average inference latency and power consumption of each image are approximately linear in the amount of computation (Multiply-Adds) [16], such that reducing computational costs helps for both improving user experience and preserving battery life. To this end, we investigate the practical inference speed of our method on an iPhone XS Max (with Apple A12 Bionic) using TFLite [1]. The single-thread mode with batch size 1 is used following [17, 43, 16]. We first measure the time consumption of obtaining the prediction with each possible length of the input sequence, and then take the weighted average according to the number of validation samples using each length. The results are shown in Figure 5. One can observe that GFNet effectively accelerates the inference of both MobileNets-V3 and ResNets. For instance, our method reduces the required latency to achieve 75.4% test accuracy (MobileNets-V3-Large) by 22% (12.7ms v.s. 16.3ms). For ResNets, GFNet generally requires $2-3\times$ times lower latency to achieve the same performance as baselines.

**Maximum input sequence length $T$ and patch size.** We show the performance of GFNet with varying $T$ and different patch sizes under the *Anytime prediction* setting in Figure 6. The figure

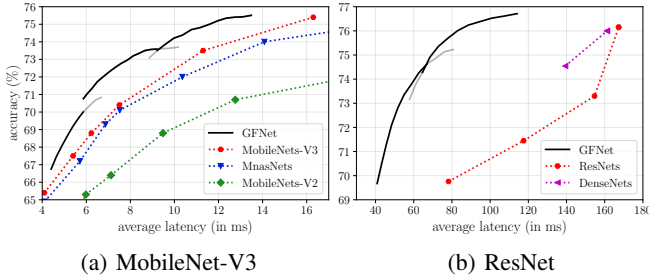
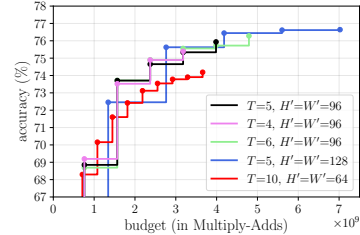

(a) MobileNet-V3  (b) ResNet

Figure 5: Top-1 accuracy v.s. inference latency (ms) on ImageNet. The inference speed is measured on an iPhone XS Max. We implement GFNet based on MobileNet-V3 and ResNet.

Figure 6: Performance of GFNet with varying $T$ and different patch sizes. Here we use ResNet-50 as the two backbones.

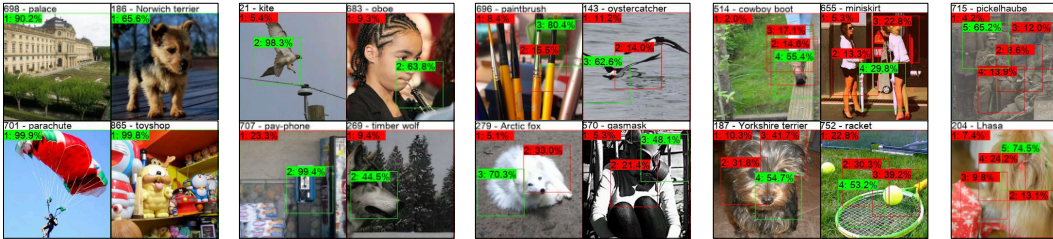

Figure 7: Visualization results of the GFNet (ResNet-50, $T=5$, $H'=W'=96$). The boxes indicate the patch locations, and the color denotes whether the prediction is correct at current step (green: correct; red: wrong). Note that $\tilde{x}_1$ is the resized input image. The indices of the steps and the current confidence on the ground truth labels (shown at the top of images) are presented in the upper left corners of boxes.

suggests that changing $T$ does not significantly affect the performance with the same amount of computation, while using larger patches leads to better performance with large computational budgets but lower test accuracy compared with smaller patches when the computational budget is insufficient.

## 4.2 Visualization

We show the image patches found by a ResNet-50 based GFNet on some of test samples in Figure 7. Samples are divided into different columns according to the number of inputs they require to obtain correct classification results. One can observe that GFNet classifies "easy" images containing large objects with prototypical features correctly at the *Glance Step* with high confidence, while for relatively "hard" images which tend to be complex or non-typical, our network is capable of focusing on some class-discriminative regions to progressively improve the confidence.

To give a deeper understanding of our method, we visualize the expected length of the input sequence E($t$) during inference under the *Budgeted batch classification* setting and the corresponding Top-1 accuracy on ImageNet, as shown in

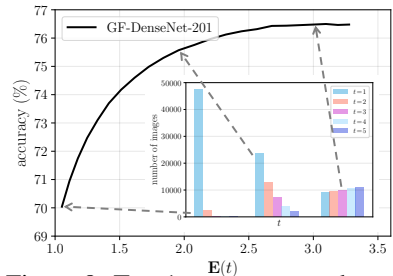

Figure 8: Top-1 accuracy v.s. the expected length of input sequence E($t$) during inference in *Budgeted batch classification*. The results are obtained based on the GFNet (DenseNet-201, $T=5$, $H'=W'=96$).

Figure 8. We also present the plots of numbers of images v.s. $t$ at several points. It can be observed that the performance of GFNet is significantly improved by letting images exit later in the *Focus Stage*, which is achieved by adjusting the confidence thresholds online without additional training.

## 4.3 Ablation Study

The ablation study results are summarized in Table 1. We first consider two alternatives of the learned patch selection policy, namely the random policy where all patches are uniformly sampled from the image, and the centre-corner policy where the network sequentially processes the whole image by first cropping the patch from the centre of the image, and then traversing the corners. The learned policy is shown to consistently outperform them. We also remove or alter the components of the GFNet architecture and the training process. One can observe that resizing the original image as $\tilde{x}_1$

Table 1: Ablation study results. For clear comparisons, we report the Top-1 accuracy of different variants with fixed length of the input sequence (denoted by $t$). The best results are **bold-faced**.

| Ablation | $t=1$ | $t=2$ | $t=3$ | $t=4$ | $t=5$ |
|---|---|---|---|---|---|
| Random Policy | 54.53% | 64.42% | 68.09% | 69.85% | 70.70% |
| Random Policy + *Glance Step* | 69.47% | 72.32% | 73.47% | 74.04% | 74.46% |
| Centre-corner Policy | 59.51% | 66.75% | 69.53% | 72.83% | 74.10% |
| Centre-corner Policy + *Glance Step* | 69.06% | 72.94% | 73.88% | 74.47% | 75.12% |
| w/o Global Encoder $f_G$ (single encoder) | 65.92% | 70.59% | 72.59% | 73.72% | 74.26% |
| w/o Regularizing CNNs (using $\mathcal{L}_{cls}$ instead of $\mathcal{L}'_{cls}$) | 68.55% | 71.87% | 73.19% | 73.52% | 73.94% |
| Initializing $f_G$ w/o $H' \times W'$ Fine-tuning | 67.81% | 72.50% | 74.12% | 75.21% | 75.56% |
| w/o the *Glance Step* (using a centre crop as $\tilde{x}_1$) | 59.14% | 66.70% | 70.91% | 73.71% | 74.70% |
| w/o Training Stage III (2-stage training) | **69.62%** | 73.39% | 74.32% | 74.97% | 75.36% |
| GFNet (ResNet-50, $T=5$, $H'=W'=96$) | 68.85% | **73.71%** | **74.65%** | **75.34%** | **75.93%** |

(*Glance Step*) and adopting two encoders are both important techniques to achieve high accuracy, especially at the first three steps. Moreover, using $\mathcal{L}'_{cls}$ helps improve the performance when $t$ is large.

## 5   Conclusion

In this paper, we introduced a *Glance and Focus Network* (GFNet) to reduce the spatial redundancy in image classification tasks. GFNet processes a given high-resolution image in a sequential manner. At each step, GFNet processes a smaller input, which is either a down-sampled version of the original image or a cropped patch. GFNet progressively performs classification as well as localizes discriminative regions for the next step. This procedure is terminated once sufficient classification confidence is obtained, leading to an adaptive manner. Our method is compatible with a wide variety of modern CNNs and is easy to implement on mobile devices. Extensive experiments on ImageNet show that GFNet significantly improves the computational efficiency even on top of the most SOTA light-weighted CNNs, both theoretically and empirically.

## Acknowledgments

This work is supported in part by the National Science and Technology Major Project of the Ministry of Science and Technology of China under Grants 2018AAA0100701, the National Natural Science Foundation of China under Grants 61906106 and 61936009, the Institute for Guo Qiang of Tsinghua University and Beijing Academy of Artificial Intelligence.

## Broader Impact

Image classification is known as a fundamental task in the context of computer vision, and has a wide variety of application scenarios, such as content-based image search, autonomous vehicles, fault detection and landmark recognition. As a flexible efficient inference framework, the proposed GFNet may help for developing resource efficient image classification systems for these applications. For example, search engines, social media companies and online advertising agencies, all must process large volumes of data on limited hardware resources, where our method can be implemented to reduce the required amount of computational resources. On mobile phones or edge devices, GFNet may also contribute to improving user experience and preserving battery life through reducing latency and the required computation. Mobile app developers or phone manufacturers may benefit from our method. On the other hand, our method also benefits environmental protection by decreasing power consumption.

Besides, in terms of the deep learning research community, our work may motivate other researchers to develop more efficient CNNs by designing more effective mechanisms to reduce spatial redundancy. Our method also has the potentials to be modified for other important vision tasks including semantic segmentation, object detection, instance segmentation, etc., which may lead to larger positive impacts.

However, GFNet suffers from the common problems of CNNs as well, such as the safety risk caused by potential adversarial attacks or the privacy risk. In addition, when improperly used, our method may also reduce the cost of criminal behaviors.

Overall, we believe that the benefits of our work to both the industry and the academia will significantly outweigh its harms.

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
