[Supplementary Material]

# Appendix: Glance and Focus: a Dynamic Approach to Reducing Spatial Redundancy in Image Classification

## A   Implementation Details

### A.1   Recurrent Networks

For RegNets [40], MobileNets-V3 [16] and EfficientNets [50], we use a gated recurrent unit (GRU) with 256 hidden units [7] in the patch proposal network $\pi$. For ResNets [14] and DenseNets [22], we adopt 1024 hidden units and remove the convolutional layer in $\pi$. This does not hurt the efficiency since here the computational cost of $\pi$ is negligible compared with the two encoders. With regards to the recurrent classifier $f_{\mathrm{C}}$, for ResNets [14], DenseNets [22] and RegNets [40], we use a GRU with 1024 hidden units. For MobileNets-V3 [16] and EfficientNets [50], we find that although a GRU classifier with a large number of hidden units achieves excellent classification accuracy, it is excessively computationally expensive in terms of efficiency. Therefore, we replace the GRU with a cascade of fully connected classification layers. In specific, at $t^{\mathrm{th}}$ step, we concatenate the feature vectors of all previous inputs $\{\bar{e}_1, \ldots, \bar{e}_t\}$, and use a linear classifier with the size $tF \times C$ for classification, where $F$ is the number of feature dimensions and $C$ is the number of classes. Similarly, we use another $(t+1)F \times C$ linear classifier at $(t+1)^{\mathrm{th}}$ step. Totally, we have $T$ linear classifiers with the size $F \times C, 2F \times C, \ldots, TF \times C$.

### A.2   Policy Gradient Algorithm

During training, the objective of the patch proposal network $\pi$ is to maximize the sum of the discounted rewards:

$$\max_{\pi} \mathbb{E}\left[\sum_{t=2}^{T} \gamma^{t-2} r_t\right], \tag{4}$$

where, $\gamma \in (0,1)$ is a pre-defined discount factor, $r_t$ is the reward for the localization action $l_t$, and $T$ is the maximum length of the input sequence. The action $l_t$ is stochastically chosen from a distribution parameterized by $\pi$: $l_t \sim \pi(l_t|e_{t-1}, h_{t-2}^{\pi})$, where we denote the hidden state maintained within $\pi$ by $h_{t-2}^{\pi}$. Here we use a Gaussian distribution during training, whose mean is outputted by $\pi$ and standard deviation is pre-defined as a hyper-parameter. At test time, we simply adopt the mean value as $l_t$ for a deterministic inference process. Note that, we always resize the original image $x$ to $H' \times W'$ as $\tilde{x}_1$ (*Glance Step*), and thus we do not have $l_1$ or $r_1$.

In this work, we implement the proximal policy optimization (PPO) algorithm proposed by [44] to train the patch proposal network $\pi$. In the following, we briefly introduce its procedure. For simplicity, we denote $\pi(l_t|e_{t-1}, h_{t-2}^{\pi})$ by $\pi(l_t|s_t)$, where $s_t$ is the current state containing $e_{t-1}$ and $h_{t-2}^{\pi}$. First, we consider a surrogate objective:

$$L_t^{\mathrm{CPI}} = \frac{\pi(l_t|s_t)}{\pi_{\mathrm{old}}(l_t|s_t)} \hat{A}_t, \tag{5}$$

where $\pi_{\mathrm{old}}$ and $\pi$ are the patch proposal network before and after the update, respectively. The advantage estimator $\hat{A}_t$ is computed by:

$$\hat{A}_t = -V(s_t) + r_t + \gamma r_{t+1} + \cdots + \gamma^{T-t} r_T, \tag{6}$$

where $V(s_t)$ is a learned state-value function that shares parameters with the policy function (they merely differ in the final fully connected layer). Since directly maximizing $L^{\mathrm{CPI}}$ usually leads to an excessively large policy update, a clipped surrogate objective is adopted [44]:

$$L_t^{\mathrm{CLIP}} = \min\left\{\frac{\pi(l_t|s_t)}{\pi_{\mathrm{old}}(l_t|s_t)} \hat{A}_t, \mathrm{clip}(\frac{\pi(l_t|s_t)}{\pi_{\mathrm{old}}(l_t|s_t)}, 1-\epsilon, 1+\epsilon)\hat{A}_t\right\}, \tag{7}$$

where $0 < \epsilon < 1$ is a hyper-parameter. Then we are ready to give the final maximization objective:

$$\underset{\pi}{\mathrm{maximize}} \quad \mathbb{E}_{x,t}\left[L_t^{\mathrm{CLIP}} - c_1 L_t^{\mathrm{VF}} + c_2 S_{\pi}(s_t)\right]. \tag{8}$$

Herein, $S_{\pi}(s_t)$ denotes the entropy bonus to ensure sufficient exploration [58, 37, 44], and $L_t^{\mathrm{VF}}$ is a squared-error loss on the estimated state value: $(V(s_t) - V^{\mathrm{target}}(s_t))^2$. We straightforwardly let $V^{\mathrm{target}}(s_t) = r_t + \gamma r_{t+1} + \cdots + \gamma^{T-t} r_T$. The coefficients $c_1$ and $c_2$ are pre-defined hyper-parameters.

In our implementation, we execute the aforementioned training process in Stage II of the 3-stage training scheme. To be specific, we optimize Eq. (8) using an Adam optimizer [28] with $\beta_1 = 0.9$, $\beta_2 = 0.999$ and a learning rate of 0.0003. We set $\gamma = 0.7$, $\epsilon = 0.2$, $c_1 = 0.5$ and $c_2 = 0.01$. The size of the mini-batch is set to 256. We train the patch proposal network $\pi$ for 15 epochs and select the model with the highest final validation accuracy, i.e., the accuracy when $t = T$. These hyper-parameters are selected on the validation set of ImageNet and used in all our experiments.

### A.3   Training Details

**Initialization.** As introduced in the paper, we initialize the local encoder $f_{\mathrm{L}}$ using the ImageNet pre-trained models, while initialize the global encoder $f_{\mathrm{G}}$ by first fine-tuning the pre-trained models with all training samples resized to $H' \times W'$. To be specific, for ResNets and DenseNets, we use the pre-trained models provided by pytorch [39], for RegNets, we use the pre-trained models provided by their paper [40], and for MobileNets-V3 and EfficientNets, we first train the networks from scratch following all the details mentioned in their papers [50, 16] to match the reported performance, and use the obtained networks as the pre-trained models. For $H' \times W'$ fine-tuning, we use the same training hyper-parameters as the training process [14, 22, 40, 50, 16]. Notably, when MobileNets-V3 and EfficientNets are used as the backbone, we fix the parameters of the global encoder $f_{\mathrm{G}}$ after initialization and do not train it any more, which we find is beneficial for the final performance of the *Glance Step*.

**Stage I.** We train all networks using a SGD optimizer [14, 22, 56] with a cosine learning rate annealing technique and a Nesterov momentum of 0.9. The size of the mini-batch is set to 256, while the L2 regularization coefficient is set to 5e-5 for RegNets and 1e-4 for other networks. The initial learning rate is set to 0.1 for the classifier $f_{\mathrm{C}}$. For the two encoders, the initial learning rates are set to 0.01, 0.01, 0.02, 0.005 and 0.005 for ResNets, DenseNets, RegNets, MobileNets-V3 and EfficientNets, respectively. The regularization coefficient $\lambda$ (see: Eq. (3) in the paper) is set to 1 for ResNets, DenseNets and RegNets, and 5 for MobileNets-V3 and EfficientNets. We train ResNets, DenseNets and RegNets for 60 epochs, MobileNets-V3 for 90 epochs and EfficientNets for 30 epochs.

**Stage II.** We train the patch proposal network $\pi$ using an Adam optimizer [28] with the hyper-parameters provided in Appendix A.2. The standard deviation of the Gaussian distribution from which we sample the localization action $l_t$ is set to 0.1 in all the experiments.

**Stage III.** We use the same hyper-parameters as Stage I, except for using an initial learning rate of 0.01 for the classifier $f_{\mathrm{C}}$. Moreover, we do not execute this stage for EfficientNets, since we do not witness an improvement of performance.

Table 2: Details of the GFNets in Figure 4 of the paper

| Backbone CNNs | GFNets |
|---|---|
| ResNets | (1) ResNet-50, $H' = W' = 96, T = 5$ |
| | (2) ResNet-50, $H' = W' = 128, T = 5$ |
| DenseNets | (1) DenseNet-121, $H' = W' = 96, T = 5$ |
| | (2) DenseNet-169, $H' = W' = 96, T = 5$ |
| | (3) DenseNet-201, $H' = W' = 96, T = 5$ |
| RegNets | (1) RegNet-Y-600MF, $H' = W' = 96, T = 5$ |
| | (2) RegNet-Y-800MF, $H' = W' = 96, T = 5$ |
| | (3) RegNet-Y-1.6GF, $H' = W' = 96, T = 5$ |
| MobileNets-V3 | (1) MobileNet-V3-Large (1.00), $H' = W' = 96, T = 3$ |
| | (2) MobileNet-V3-Large (1.00), $H' = W' = 128, T = 3$ |
| | (3) MobileNet-V3-Large (1.25), $H' = W' = 128, T = 3$ |
| EfficientNets | (1) EfficientNet-B2, $H' = W' = 128, T = 4$ |
| | (2) EfficientNet-B3, $H' = W' = 128, T = 4$ |
| | (3) EfficientNet-B3, $H' = W' = 144, T = 4$ |

The input size $(H', W')$, the maximum input sequence length $T$ and the corresponding encoders used by the GFNets in Figure 4 of the paper are summarized in Table 2. Note that we always let $H' = W'$.

|  |  |  |
|---|---|---|
| (a) MobileNet-V3 | (b) RegNet-Y | (c) EfficientNet |

|  |  |
|---|---|
| (d) ResNet | (e) DenseNet |

Figure 9: The performance of $H'{\times}W'$ fine-tuned models. The results of *Budgeted batch classification* are presented. Note that, the performance of the *Glance Step* is mainly determined by the low-resolution fine-tuning, and thus this fine-tuning is an important component of the proposed GFNet framework.

## B  Additional Results

### B.1  Effects of Low-resolution Fine-tuning

As mentioned in the paper, our method initializes the global encoder $f_G$ by first fine-tuning the pre-trained models with all training samples resized to $H'{\times}W'$. An interesting observation is that the low-resolution fine-tuning improves the computational efficiency by itself. The performance of the fine-tuned models compared with baselines and GFNets is reported in Figure 9. It is important that the improvements achieved by these fine-tuned models are actually included in our GFNets, since the performance of the *Glance Step* is mainly determined by the low-resolution fine-tuning. On the other hand, our method is able to further improve the test accuracy with the *Focus Stage*, and adjust the average computational cost online.

### B.2  Comparisons with MSDNet in *Budgeted Batch Classification*

Figure 10: Performance of GFNet (based on DenseNets) versus MSDNet [20] under the *Budgeted Batch Classification* setting.

The comparisons of DenseNet-based GFNets and MSD-Nets [20] under the *Budgeted Batch Classification* setting are shown in Figure 10. Following [20], here we hold out 50,000 images from the training set as an additional validation set to estimate the confidence thresholds, and use the remaining samples to train the network (note that we use the entire training set in Figure 4 (e) of the paper). One can observe from the results that GFNet consistently outperforms MSDNet within a wide range of computational budgets. For example, when the budgets are around $1 \times 10^9$ Multiply-Adds, the test accuracy of our method is higher than MSDNet by approximately $2\%$. GFNet is shown to be a more effective adaptive inference framework than MSDNet. In addition, in terms of the flexibility of GFNet, its computational efficiency can be further improved by applying state-of-the-art CNNs as the two encoders.