[Reviews · NeurIPS 2020]

Review 1

Summary and Contributions: This paper propose a dynamic approach to crop and classify images. As a classifier, it auto-zoom-in the image to see more clearly, which equals to working on a high resolution image.

Strengths: 1) The method works well on imageNet-1k, perhaps because most images in imageNet-1k contains a salient object, rarely categories are scenarios. But this method seems to be able to focus on the important and discriminatve part of the object, such as dog head in Figure 1. 2) It has beated MSDNet consistently across range of FLOPS. Therefore I think this method really make something difference. 3) The method has been tested on IPhone, which verifies the effectiveness. 4) The two stages of detection-and-classify pipeline has been widely used in face recognition, which also validates this kind of method would be successful.

Weaknesses: I think some comparsions are necessary. 1) Maybe regressing a bounding box for the image and then classify it yield to similar results, under the condition of some images are given localization bounding box in imageNet-1k. 2) I think the upper-bound of this auto-zoom-in method is directly feeding in high resoution images. The related comparsions are meaningful. 3) I feel the design of RL and iterative process is too complex, and unncessary, usually.

Correctness: Yes.

Clarity: Yes.

Relation to Prior Work: Yes.

Reproducibility: No

Additional Feedback: This experiment design is very complex but the results seems promising. I think the authors should make the code and model public once acceptance of this paper. I feel this paper is a good technical paper and it is slightely above the acceptance bar.


Review 2

Summary and Contributions: The paper seeks to extend the line of work on recurrent models of visual attention (RMVA). In particular, it shows for the first time that RMVAs can yield much faster inference in the standard image classification scenario (ImageNet) with respect to state-of-the-art models (ResNet, DenseNet, etc) while preserving sota accuracies. The modifications to the RMVA architecture mostly seem minor conceptually (e.g. the use of a standard backbone network as the glimpse network instead of the original tiny network; allowing early exit from the recurrence), although they use a very different reinforcement learning scheme (which is surprsingly not called out in their summary of differences in Section 2). But there are many engineering details, which are apparently the key to getting things to work. There is a very detailed and well-done evaluation against state-of-the-art systems, including an informative ablation study. Given the applicability to image classification, if the claims in this paper held, I expected forms of GFNets to become standard for attention-based classification. However, some details in the results make me question where the gains are actually coming from (see below).

Strengths: + Clean, sensible and mostly simple extension of the original RMVA architecture. The main non-obvious change is in replacing the reinforcement learning algorithm, which was originally REINFORCE, with PPO. + Extensive evaluation showing sota accuracies with significant speedups in the standard image classification scenarios. This is the first work I've seen in the RMVA line that head-on addresses the issue of improving standard image classification. Note RMVA itself was only evaluated on MNIST. Follow-up work from Ba et al looked at image captioning and included, e.g., the "innovation" in the current paper of using a serious convnet as the "glimpse" network. But surprisingly, haven't seen anyone attempt the obvious task of improving base image classification with RMVAs.

Weaknesses: Post rebuttal edit: The extra experiments comparing to random did convince me that GFNet does something beyond random. But I'm still not convinced that GFNets are particularly smart at glancing. Note (from table 1 of the rebuttal) for instance that to reach sota accuracy, it looks like a fovea/glance of size 1/n of the original window seems to need ~n steps. To me this seems that glancing barely pays for itself. In fact, if you replaced random foveation with deterministic uniform coverage of the image, you may have done better. I'll bump up my score, but I really hope you can address the above issues so that the regime where gfnets give usable performance improvements is very clear. When should gfnets be used? ---------------- The trickiest part of getting RMVA-style models to work is training the recurrent localizer (the "visual attention" module), usually via reinforcement learning. The magic of the RMVA approach is when the policy makes economical choices of locations within the image to focus on. In this regard, Figure 7 was reassuring. I initially just had a minor question about why the authors chose to replace REINFORCE with PPO, and requests to get a lot more insight into the RL training. For instance, what was the impact of the clipped surrogate, what was the impact of the entropy bonus, how long did training take, etc. However, on seeing the ablation study, I was surprised to see that a random policy (row 1 of table 1) seems to consistently do within 1.3% points of accuracy relative to the learned policy! This implies that the policy is not doing very much: basically a system that randomly samples locations and exits early should get almost all the benefits that GFNet gets. It seems that GFNet is getting most of its benefits from early exiting. And in fact, comparing with the early-exit version of DenseNets (i.e., MSDNets), it seems that your gains are very similar (compare Fig 4e of this paper to Fig 5(left) of https://arxiv.org/pdf/1703.09844.pdf. One way to confirm this is to make your window size (H', W') much smaller than the original image size, so that random cannot do well on it. Does GFNet still do well? As of now, this paper provides an alternate (RNN-based) way to implement early-exit models, which is interesting, but not as high-impact as showing how to get visual attention to work on ImageNet.

Correctness: Yes, the paper makes justifiable modifications to the base RMVA system, has plenty of the right kind of detail describing these and has good evaluation methodology. The fatal flaw as of now, however, is the claim in the paper that it is dynamically identifying "class-discriminative" regions of individual images. The good performance of randomly selected baseline regions relative to their system does not seem to support this claim.

Clarity: yes

Relation to Prior Work: yes

Reproducibility: Yes

Additional Feedback: I was very excited to read this paper because it brought RMVA to the real world. I was therefore quite taken aback to notice the performance of the random policy. Perhaps you could add an experiment with a much smaller foveation window so that random fails and the value of GFNet is clear? Also, compare to something like MSDNet to show that your benefit s are not "just" coming from early exit, but also due to intelligent foveation/visual attention.


Review 3

Summary and Contributions: The paper presents a glance-and-focus design, where the proposed framework glances a low-resolution image at first and progressively focus on high-resolution patches until it achieved the prediction confidence.

Strengths: In order to reduce the spatial redundancy, the proposed framework efficiently identify the regions of interest with low-resolution image, and progressive identify and process high-resolution image patches. According to the confidence of prediction, the framework dynamically adjusts the computation complexity. It's interesting to see a solution to reduce the computation complexity through the integration of low/high-resolution processing, reducing spatial redundancy, and dynamic/conditional execution. Evaluations on imagenet have illustrated the superior accuracy-efficiency tradeoff compared to other STOA lightweight CNNs and anytime prediction solutions.

Weaknesses: It seems that, in the evaluation part, the paper does not compare with recent efforts on conditional computation that focus on reducing spatial redundancy (e.g., [1][2]). Although, as far as I know, none of them combine multi-resolution with spatially adaptive computation, it's good to see the comparison of end-to-end latency and accuracy. [1]Figurnov, Michael, et al. "Spatially adaptive computation time for residual networks." Proceedings of the IEEE Conference on Computer Vision and Pattern Recognition. 2017. [2]Ren, Mengye, et al. "Sbnet: Sparse blocks network for fast inference." Proceedings of the IEEE Conference on Computer Vision and Pattern Recognition. 2018.

Correctness: The claims and method looks correct to me.

Clarity: The paper is well written. The motivation is delivered clearly, and the proposed framework is illustrated well with the figures. However, authors can define the averaged budget and explain why the curves are across in Figures 4 and 5 in more detail.

Relation to Prior Work: As mentioned above, authors haven't enough discussion and comparison with previous works on spatial adaptive computation (e.g., [1][2]). It will be better if authors could compare the proposed framework with those work empirically.

Reproducibility: Yes

Additional Feedback:


Review 4

Summary and Contributions: Edit post rebuttal: The authors have satisfactority answered my questions and concerns in the rebuttal. After having read the other reviews, I stick to my initial assessment of accepting the paper. The authors present Glance and Focus net which at its core posseses a recurrent mechanism to do image classification. In the glance step the network does image classification using standard pretrained networks albeit with an image of a reduced resolution. In the focus step, a patch proposal network crops out interesting parts of the image to focus on and make a prediction with a higher confidence. The authors describe the benefit of their approach with respect to compute vs accuracy trade-off on Imagenet benchmarks.

Strengths: 1. The approach proposed is orthogonal to other approaches tacking the problem of efficiency of neural networks., i.e. process the image sequentially as opposed to adapting the architecture. 2. The patch proposal network is novel contribution. The architecture and training strategy to mitigate the non-differentiability of the crop process can be useful to other image processing tasks. 3. Thorough ablation study on the usefulness of each component in the approach. Also the visualizations illustrate the network is working as intended. 4.Ability for the approach to be applied to any backbone and validating that the approach is agnotic to the choice of a particular network type, including illustrating improvements over SOTA efficientnets and reg-nets. 5. Illustrating the usefullness on hardware (iPhone) is useful to practioners.

Weaknesses: 1.The authors have done a good job with placing their work appropriately. One point of weakness is insufficient comparison to approaches that aim to reduce spatial redudancy, or make the networks more efficient specifically the ones skipping layers/channels. Comparison to OctConv and SkipNet even for a single datapoint with say the same backbone architecture will be valuable to the readers. 2. The authors need to show a graph showing the plot of T vs number of images, and Expectation(T) over the imagenet test set. It is important to understand whether the performance improvement stems solely from the network design to exploit spatial redundancies, or whether the redudancies stem from the nature of ImageNet, ie., large fraction of images can be done with Glance and hence any algorithm with lower resolution will have an unfair advantage. Note, algorithms skipping layers or channels do not enjoy this luxury. 3. The authors should add results from [57] and discuss the comparison. Recent alternatives to MSDNets should be compared and discussed. 4. Efficient backbone architectures and approaches tailoring the computation by controlling convolutional operator have the added advantage that they can be generally applied to semantic (object recognition) and dense pixel-wise tasks. Extension of this approach, unlike other approaches exploiting spatial redundancy to alternate vision tasks is not straightforward. The authors should discuss the implications of this approach to other vision tasks.

Correctness: The authors have not supplemented their work with code. The url provided is inactive.The claims and method are sensible, and they have followed best practices from [57] in their evaluation.

Clarity: Yes, the authors have done a good job with writing the paper. The arguments flow well and I could not catch any typos.

Relation to Prior Work: The authors do a good job discussing reletvant prior work. I found several parts of the work to be similar in sprit to SkipNet. The authors should enlist the similarities and differences to skipnet as they did for recurrent visual attention models.

Reproducibility: Yes

Additional Feedback: Overall, I like the ideas presented in the paper and believe it can spur new directions. The quality of the paper will be substantially improved if the weakness are addressed appropriately.

[Author Response · NeurIPS 2020]

#R1# **Using Bounding Boxes.** In fact, we considered a more general assumption where bounding boxes are absent, in
terms that annotating datasets with localization boxes is usually expensive and time-consuming in realistic applications.
We agree that it is a nice idea to exploit the bounding boxes of ImageNet, and are happy to explore it in GFNet.
#R1# **Upper Bound of Performance.** Thanks for the suggestion. The accuracy of GFNets will approach the models
receiving the whole images (e.g., 224x224) when using larger patches (e.g., with DenseNet-121, the gaps are 0.6%/0.1%
for 96x96/128x128). We will add more comparisons on this point in our revision.
#R1# **Complex Training Process.** Given that no ground-truth bounding box is provided, we believe that RL is crucial
to search for a superior patch selection strategy. The RL part is not complex since we simply use an off-the-shelf PPO
algorithm. The iterative process is indeed not indispensable, but in experiments, it improves the accuracy (e.g., $\sim 0.5\%$
with MobileNet-V3) compared with training all components simultaneously. We will make these clear in revision.
11
#R1# **Code.** We will release all the code and pre-trained models upon the acceptance of this paper.
Table 1: Results using 32x32 (left) and 64x64 (right) patches. Due to time limits, more results will be added in revision.

| 32x32 Patches | $t=1$ | $t=2$ | $t=3$ | $t=4$ | 64x64 Patches | $t=1$ | $t=2$ | $t=3$ | $t=4$ | $t=5$ | $t=6$ | $t=7$ | $t=8$ | $t=9$ | $t=10$ |
|---|---|---|---|---|---|---|---|---|---|---|---|---|---|---|---|
| Random Policy | 41.65% | 45.31% | 47.66% | 49.10% | Random Policy | 60.88% | 64.37% | 66.53% | 67.87% | 68.90% | 69.52% | 70.05% | 70.41% | 70.73% | 70.94% |
| GF-ResNet-50 | **41.93%** | **52.04%** | **55.32%** | **57.49%** | GF-ResNet-50 | **61.03%** | **68.24%** | **70.18%** | **71.61%** | **72.36%** | **73.10%** | **73.64%** | **73.84%** | **74.07%** | **74.27%** |

#R2# **Selection of RL Algorithm.** We use PPO since it is shown to outperform REINFORCE in terms of effectiveness,
efficiency and stability in its original paper. We will study the impacts of its components in our revision.
#R2# **Performance of Random Policy.** Thanks for pointing out this issue. There are three reasons why random
policy performs well. First, the sequential classification task on ImageNet is not that difficult for random policy. It has
been proven in [1] that ImageNet-trained CNNs are strongly biased towards recognizing image textures rather than
shapes. Even random patches can capture textures (local patterns) well, leading to acceptable performance. Second, in
ablation study, we use relatively large patches of 96x96 (approaching 1/4 of 224x224 images), which are very likely
to contain some class-discriminative regions even with random sampling. As shown in Table 1, GFNet outperforms
random policy by 8% and 4% when using 32x32 and 64x64 patches. Third, for fair comparison, we also augment the
random policy with the *Glance Step*, which contributes to an excellent preliminary prediction. In addition, it is actually
challenging for GFNet to learn to identify class-discriminative regions since we considered a very general setting where
no localization annotation (e.g., bounding boxes) is available. We admit that there may still exist space to design better
RL algorithms, and will focus on this point in the future. ([1] Geirhos R, et al. ImageNet-trained CNNs are biased
towards texture; increasing shape bias improves accuracy and robustness. In ICLR, 2019.)
28
#R2# **Comparisons with MSDNet.** As suggested, we compare GFNet with MSDNet in Figure 1.
30

Figure 1: Performance of GFNets and baselines. OctConv-ResNets are not presented as they use a different network architecture from us (a pre-act version). We will implement pre-act GF-ResNets and present comparisons in revision.

Figure 2: Top-1 acc. v.s. E($t$). We show the numbers of images using different values of $t$ in several points.

#R3# **More Baselines.** We compare SACT with GFNet in Figure 1. Since SBNet is proposed for 3D object detection,
we do not directly compare it with ours. We will thoroughly discuss the relations of GFNet to the two works in revision.
#R3# **Clarity.** Thanks for the suggestion. The averaged budget refers to the mean computational cost of each test
sample. Each curve in Figure 4&5 of the paper corresponds to an individual model. The curves are across since each
model has the highest efficiency only within a certain range of computational budgets. We will make these points clear.
36
#R4# **More Baselines.** We compare OctConv, SkipNets and RANet with GFNet in Figure 1. In fact, We believe that
OctConv and SkipNet are orthogonal techniques with our method since they can be used as CNN backbones in GFNet.
#R4# **Value of E($t$).** As suggested, we show the graph of E($t$) v.s. accuracy in Figure 2. We also present the plots of $t$
v.s. number of images. One can observe that the performance of GFNet is significantly improved by letting images exit
later in the *Focus Stage*, which is realized via adjusting the confidence thresholds online (without additional training).
#R4# **Implications to Other Vision Tasks.** Although we only focus on the most general classification task in this
paper, we note that it is causable to extend GFNet to other vision tasks. For example, in objective detection, we can
obtain the preliminary prediction using low-resolution inputs (*Glance*), and then *focus* the computation on "important"
high-resolution local regions to sequentially find all the objects. We will add these discussions in revision.
#R4# **Relation to SkipNet.** Indeed, GFNet is similar to SkipNet in terms of the dynamic architecture. However, we
focus on reducing spatial redundancy, while they adaptively skip unnecessary layers. We will include more discussions.

[Meta-Review · NeurIPS 2020]

Four knowledgeable referees support acceptance for the contribution; they like the authors' novel idea of Glance and Focus net and ImageNet scale evaluations showing superior accuracy-efficiency tradeoff against state-of-the-art-baselines. Please make it sure to properly include and discuss missing references and experimental comparisons, as promised in the rebuttal. [1] (and related works) is also related from another perspective and should be properly discussed in related work - it is using early-exist inference methods via RL but for missing data (if image patches in different resolutions are understood as 'missing' features, then they share the same spirit). [1] Shim et al. Joint Active Feature Acquisition and Classification with Variable-Size Set Encoding. NeurIPS 2018